# The Role of Ketone Bodies in Treatment Individualization of Glioblastoma Patients

**DOI:** 10.3390/brainsci13091307

**Published:** 2023-09-11

**Authors:** Corina Tamas, Flaviu Tamas, Attila Kovecsi, Georgiana Serban, Cristian Boeriu, Adrian Balasa

**Affiliations:** 1Doctoral School, “George Emil Palade” University of Medicine, Pharmacy, Science and Technology, 540142 Targu Mures, Romania; corina.hurghis@umfst.ro (C.T.); georgiana.serban94@yahoo.com (G.S.); 2Neurosurgery Department, Emergency Clinical County Hospital, 540136 Targu Mures, Romania; adrian.balasa@yahoo.fr; 3Department of Neurosurgery, “George Emil Palade” University of Medicine, Pharmacy, Science and Technology, 540142 Targu Mures, Romania; 4Department of Morphopathology, “George Emil Palade” University of Medicine, Pharmacy, Science and Technology, 540142 Targu Mures, Romania; kovecsiattila@gmail.com; 5Department of Morphopathology, Emergency Clinical County Hospital, 540136 Targu Mures, Romania; 6Department of Anesthesiology and Intensive Care, Emergency Clinical County Hospital, 540136 Targu Mures, Romania; 7Department of Emergency Medicine, George Emil Palade University of Medicine, Pharmacy, Science, and Technology, 540142 Targu Mures, Romania; cristian.boeriu@umfst.ro; 8Department of Emergency Medicine, Emergency Clinical County Hospital, 540136 Targu Mures, Romania

**Keywords:** astrocytoma, glioblastoma, glucose, ketones, metabolism, diet

## Abstract

Glioblastoma is the most common and aggressive primary brain tumor in adults. According to the 2021 WHO CNS, glioblastoma is assigned to the IDH wild-type classification, fulfilling the specific characteristic histopathology. We have conducted a prospective observational study to identify the glucose levels, ketone bodies, and the glucose-ketone index in three groups of subjects: two tumoral groups of patients with histopathological confirmation of glioblastoma (9 male patients, 7 female patients, mean age 55.6 years old) or grade 4 astrocytoma (4 male patients, 2 female patients, mean age 48.1 years old) and a control group (13 male patients, 9 female patients, mean age 53.9 years old) consisting of subjects with no personal pathological history. There were statistically significant differences between the mean values of glycemia (*p* value = 0.0003), ketones (*p* value = 0.0061), and glucose-ketone index (*p* value = 0.008) between the groups of patients. Mortality at 3 months in glioblastoma patients was 0% if the ketone levels were below 0.2 mM and 100% if ketones were over 0.5 mM. Patients with grade 4 astrocytoma and the control subjects all presented with ketone values of less than 0.2 mM and 0.0% mortality. In conclusion, highlighting new biomarkers which are more feasible to determine such as ketones or glucose-ketone index represents an essential step toward personalized medicine and survival prolongation in patients suffering from glioblastoma and grade 4 astrocytoma.

## 1. Introduction

Glioblastoma (GBM) is the most common and aggressive primary brain tumor in adults accounting for up to 45.2% of the primary cerebral malignancies [1,2,3,4,5,6]. The Central Brain Tumor Registry of the United States reports an average annual incidence of 3.19/100,000 people, while the United Kingdom Office of National Statistics reports a doubling of the number of cases from 2.4 to 5.0/100,000 between the years 1995 and 2015, with the current numbers having increased from 983 to 2531 cases per year [7,8,9]. The incidence of GBM increases with age, reaching a peak among individuals between 75 and 84 years old, with a higher prevalence in men (1.57% more than in women) [6,8,10,11]. 

The previous “World Health Organization Classification of Tumors of the Central Nervous System” (WHO CNS), 2016, based on histopathological diagnosis, used the term glioblastoma, which is divided into three subclasses: Isocitrate Dehydrogenase (IDH) mutant (10%), IDH wild-type (90%), and IDH with not otherwise specified (NOS), each of which presents with a completely different biology and prognosis [12]. According to the WHO CNS 2021 classification, the term glioblastoma is assigned only to the IDH wild-type subclass, fulfilling the specific histopathological characteristics of diffuse astrocytoma but with one or more genetic modifications (Telomerase reverse transcriptase-TERT promoter mutation, chromosome 7 or chromosome 10 damage (+7/−10), or Epidermal Growth Factor Receptor-EGFR gene amplification). IDH mutant astrocytomas are considered one single subtype of varying degrees (WHO 2,3, or 4). The presence of the homozygous deletion Cyclin-dependent Kinase Inhibitor-CDKN2A/B without histopathological findings of necrosis or microvascular proliferation defines WHO grade 4 astrocytoma (ASTRO G4) [13].

Most studies focus on the mechanisms of tumor cell invasion into the brain’s microenvironment (Rho GTPases, Casein Kinase 2, and Ephrin receptors as major invasion factors) [14,15,16]. Recent studies have highlighted the reprogramming process of the cellular metabolism, which has a definitive role in preparing the cellular microenvironment for tumor invasion [17,18,19]. One of the defining characteristics of tumor development at the bioenergetic level is the ability of tumor cells to exploit the glycolytic metabolism independent of the presence of oxygen, a phenomenon known as the Warburg effect [5,6,20,21,22]. Many recent studies have questioned the possibility of using other energy sources such as ketone bodies (KBs) by GBM to generate energy [23,24,25].

Fatty acids and glucose are metabolized to acetyl coenzyme A (acetyl-CoA) inside the hepatocyte mitochondria. Acetyl-CoA enters the citric acid cycle by condensation with oxaloacetate. Glycolysis produces pyruvate, which is a precursor of oxaloacetate. If there is a significant decrease in glycolysis, oxaloacetate is preferentially used in the process of gluconeogenesis, becoming unavailable for condensation with acetyl-CoA produced through the degradation of fatty acids. In this case, acetyl-CoA deviates from the citric acid cycle to the formation of KBs (Figure 1) [22,26,27,28].

KBs are made up of three molecules: 3-β-hydroxybutyrate, acetoacetate, and acetone. 3-β-hydroxybutyrate results from the reduction of acetoacetate at the mitochondrial level and is the main transporter of energy from the liver to other tissues, of which the brain is the most important. Most tissues can use fatty acids as a source of energy during periods of severe hypoglycemia. The brain does not benefit from this adaptive mechanism and therefore, KBs are an essential alternative source of energy [29,30,31,32,33,34].

## 2. Materials and Methods

### 2.1. Aims and Scope

By using commercially available kits, we aimed to highlight the differences in blood glucose levels and KB values from the peripheral blood between three groups of patients: two tumoral groups with GBM/ASTRO G4 and a control group of healthy subjects, without influencing their diets. 

By analyzing the differences between these groups, we aimed to determine if it is possible to use KBs and the glucose-ketone index (GKI) as prognostic factors of tumoral aggression.

### 2.2. Patients

This prospective observational study was conducted in the Neurosurgery Department of the Emergency Clinical Hospital of Targu Mures between January 2021 and June 2022 in accordance with the Declaration of Helsinki. The included patients were adults (>34 years old) who provided informed consent. The protocol of this study was approved by the Hospital’s Ethics Committee. Three groups of subjects were included in the study: two groups of patients with histopathological confirmation of GBM or ASTRO G4 and a control group of subjects without a personal history of malignant pathologies. 

Patients who were on a certain diet (such as ketogenic diets or similar) or had dietary restrictions and patients suffering from diabetes or other metabolic diseases were excluded. From the tumoral groups, patients who underwent biopsy or partial resection, and patients presenting with a Karnofski Performance Score (KPS) less than 80 were excluded. From the control group, subjects with a known personal pathological history such as metabolic diseases, benignant or malignant non-glial brain tumors, or systemic cancer were excluded [35].

### 2.3. Parameters Measured

In all of the three groups, a jeun glycemia and KBs (3-β-hydroxybutyrate) from the peripheral blood were measured using available commercial kits (Medical Device NOVA PRO GLU KET CONTROL, NOVA BIOMEDICAL, Product Code 47292, Category Code W0101060108). Fasting blood sampling was performed in the early morning of the second day after admission.

Based on the glycemic values and KB levels, the GKI was calculated. By determining the GKI, a single value that expresses the relationship between glucose (major fermentable tumor fuel) and KBs (non-fermentable fuel) was obtained. As most commercial kits express blood glucose in mg/dL and ketones in mM (including the kits used in this study), glucose units were converted to mM using the following formula [36]: GKI (mM) = [Glucose (mg/dL)/18.016 (g × dL/moL)]/Ketone (mM) 

The weight and height of the included patients were also considered, and the body mass index (BMI) was calculated according to the following formula: weight (kg)/[height (m)]^2^.

### 2.4. Assessment of the Clinical Condition 

The Motor Assessment Scale was used to assess motor deficits and the Glasgow Coma Scale was used to assess consciousness [37,38]. The KPS was used to assess the functional status of patients both before and after the surgery.

### 2.5. Neuroimaging Evaluation

Preoperative magnetic resonance imaging (MRI) scans and immediate postoperative cranial computer tomography (CT) scans were performed for all patients. Follow-up MRIs were performed every 3 months during the patient’s lifetime. Tumor mass was measured based on the following formula: (maximum axial diameter × maximum coronal diameter × maximum sagittal diameter)/2 [39]. Perilesional edema has been defined as the T2 hypersignal area surrounding the tumor. The size of the edema was estimated based on the ratio of the minimum and maximum distances from the edge of the tumor to the outer edge of the edema on the axial scans.

The histopathological diagnosis could be suspected after analyzing the imaging aspects of the tumors: GBM was characterized by peripheral contrast enhancement and the central hyposignal area in the T1C+ sequence (Figure 2a–c). ASTRO G4 is sometimes difficult to categorize based on imaging aspects. It is characterized by the heterogeneity of contrast enhancement in the T1C+ sequence and a hypo/isosignal in the T1 sequence. Figure 2d–f shows the MRI aspects of a patient from the tumoral group with a confirmed histopathological diagnosis of ASTRO G4; the T1C + hyposignal area represents the area of tumoral tissue and not the area of perilesional edema.

### 2.6. Specific Medical Management

All patients included in the tumoral groups received dexamethasone at a dose of 4 to 16 mg per day according to the current treatment protocols. Preoperative medication was administered for between 1 and 5 days. Patients who survived for more than 1 month received postoperative chemotherapy and radiation therapy in accordance with the STUPP protocol [40]. 

### 2.7. Surgical Management

All surgical interventions were performed under general anesthesia. The surgical approaches were guided by the neuronavigation system (Curve 2.1; Brainlab, 81829 Munich, Germany) allowing us to perform minimal invasive craniotomies centered on the tumor’s locations. In tumors located near eloquent areas, the image injection option of our surgical microscope (Captiview; Leica Microsystems, 35578 Wetzlar, Germany) was used in conjunction with the neuronavigation system (Figure 3). 

The trajectories of approach were chosen to be as short as possible by using transsylvian or transsulcal approaches while simultaneously avoiding eloquent areas. The use of retractors was avoided; instead, “dynamic retraction” technique described by Spetzler et al. [41] was used, paying significant attention to sulcal and fissure dissections to minimize the surgical sacrifice of the brain parenchyma. 

Total resection (considered over 90%) was performed under the operating microscope (Leica Microsystems, 35578 Wetzlar, Germany). The extent of the resection was assessed by the main operator and an experienced radiologist. 

### 2.8. Histopathological Analysis

The histopathological diagnosis was established within the Pathological Anatomy Department of our institute in accordance with the WHO 2016 and 2021 classification standards for tumors of the central nervous system.

### 2.9. Statistical Analysis

Statistical analysis included elements of descriptive statistics (mean, median, and standard deviation) and inferential statistics. The Shapiro–Wilk test was applied to determine the distribution of the analyzed data series. 

The t-Student parametric test for unpaired data was applied to compare means and the non-parametric Mann–Whitney test was applied to compare medians. The significance threshold value chosen for p was 0.05 with a confidence interval of 95%. Statistical analysis was performed using the GraphPad Prism trial version utility.

## 3. Results

### 3.1. General Clinical Features

Out of a total of 22 patients with brain tumors, 27.3% had ASTRO G4. IDH mutation was present in 100% of ASTRO G4 patients. Men accounted for 59.1% of the patients and the mean age was 56.3 years (range: 34 to 74 years). 

The control group had similar characteristics to the tumor groups: 59.1% of the subjects were male, with an average age of 53.9 years (range: 36 to 78 years). There was no statistically significant difference in the mean age between the three groups (*p* ˃ 0.05 using the t-Student test, Table 1).

### 3.2. Specific Clinical Features

In the tumor groups, the average BMI (body-mass index) was 28.1%, the average GKI was 38.7%, and the average KB value was 0.2%. The median time interval from the onset of symptoms to the diagnosis of GBM or ASTRO G4 (cranial MRI examination) was 6 weeks. Headache was the main onset sign, present in 90.1% of patients, followed by motor deficit and confusion in 45.4% and 40.9% of cases, respectively. 

The control group showed an average BMI of 28.0%, an average GKI of 16.8%, and an average KB value of 0.08%. Regarding KBs, the values were less than 0.2 mM in patients with ASTRO G4, similar to the control group. 

There were no statistically significant differences (*p*-value ˃ 0.05; CI 95%) in the mean height, weight, and BMI between the three groups of subjects as per the t-Student test (Table 1). The mean values of glycemia, KBs, and GKI showed statistically significant differences between the three groups (*p*-value ˂ 0.05 using the Mann–Whitney test, Table 1).

### 3.3. Tumor Characteristics

Most tumors were located in the left frontal lobe both for GBM (54.5%) and ASTRO G4 (50% of the total number of patients). The maximum/minimum diameter was 68/24 mm for GBM and 54/21 mm for ASTRO G4, and the maximum/minimum diameter of the area of perilesional edema was 60/0.2 mm for GBM and 40/0.1 mm for ASTRO G4 (Table 2).

### 3.4. Histopathological Features

Tumor tissue consisted of atypical glial tumor cells with increased mitotic activity and variable cellularity with infiltrative character. No myxoid character or microcyst formation was noted. Tumoral cells had rounded or elongated hyperchromic nuclei with variable pleomorphism and fine, eosinophilic fibrillar processes. Tumor cells with marked pleomorphism were characterized by larger nuclei with lobed, vesicular character, sometimes with bizarre shapes. Foci of necrosis have been identified in all cases, sometimes with palisading of the surrounding nuclei, and microvascular hyperplasia with hyperplastic endothelial cells, often with the presence of “glomeruloid” bodies (Figure 4, Table 2) [42].

### 3.5. Postoperative Death

The 3-month death rate was 0.0% in patients with ASTRO G4 and 43.75% in patients with GBM. At 1 month, the death rate was 12.5% in patients with GBM. The rate of postoperative complications was 16.7% in patients with ASTRO G4 and 18.75% in those with GBM. Postoperative complications have not led to death (Table 3).

## 4. Discussion

Due to the low survival rate of patients with GBM/ASTRO G4, there is an urgent need for adjuvant therapies that increase survival and quality of life. In this context, cellular metabolism, especially glucose and KB metabolism, represents a therapeutic target and a broad topic of research [10,26,30,43,44].

KBs play essential roles in various metabolic pathways such as β-oxidation (Fatty Acid Oxidation), the biosynthesis of sterols, the tricarboxylic acid cycle, de novo lipogenesis, and gluconeogenesis [45,46,47]. They are a vital alternative for fueling the brain during periods of nutrient deprivation. KBs are mainly produced inside the liver from acetyl-CoA and are transported to extrahepatic tissues for terminal oxidation [22,26,29]. Normally, the blood levels of KBs are situated below 0.5 mM. Values between 0.5 and 1.0 mM are considered slightly higher, while values between 1.0 and 3.0 mM are considered moderately high [18,27,48,49].

The current treatment protocol for GBM and malignant astrocytomas consists of surgery, followed by radiotherapy and chemotherapy (temozolomide) [40,50,51,52]. However, the average survival duration is still less than 15 months, and the 5-year survival is below 10% [6,8,10,11]. These patients also have an increased risk of suicide, possibly due to the poor prognosis of this pathology and because of treatment-related side effects, such as mood-altering steroids [53]. Establishing new treatment regimens based on different peripheral markers that are easy to determine from the patient’s peripheral blood, such as KBs, and introducing adjuvant therapies based on these parameters, such as ketogenic diets, and thus essentially individualizing treatments, may result in an increase in the survival rate. In this context, tumoral cellular metabolism may be a new therapeutic target that warrants attention.

Most studies have revealed that brain tumor cells are dependent on glucose for survival and KBs cannot be used effectively as alternative fuels [29,46]. Therefore, this “metabolic management area,” defined by decreasing the blood glucose levels and increasing KB levels, may result in the improvement of the survival rate in patients suffering from high-grade malignancies [6,33,36,54]. It is well known that during physical exercise, fasting, carbohydrate restriction, or insulin deficiency, KB levels increase, and even if ketoacidosis is a pathological condition with serious repercussions, mild ketonemia can have beneficial effects in cancer [55,56,57]. Unfortunately, in GBM, this theory may not be applicable [26,58].

Certain oncogenic mutations such as Phosphatidylinositol-3-kinase (PI3K)/Protein kinase B (AKT)/Mammalian target of rapamycin (mTOR) significantly influence GBM metabolism by promoting the use of glucose as an energy substrate and promoting the synthesis of FAs [14,20,26,46,59]. These mutations induce the “glucose dependence” of tumor cells, which would be directly targeted by glucose deprivation [6,23,29]. Unfortunately, GBM is characterized by high heterogeneity and has even raised the hypothesis that it could use FAs as a substrate for generating new tumor blocks [29,60]. Metabolic reprogramming takes place, with a defining role in all stages of GBM development. Due to this high individualized heterogeneity of GBM, it is very difficult to establish certain easily determinable biological markers that can predict the degree of tumor aggressiveness, the response to oncological treatment, and even the need to administer adjuvant treatments, such as ketogenic diets [21,30,31,61].

In this prospective study, we explored the possibility of using KBs and GKI for predicting tumor aggressiveness in patients with GBM or ASTRO G4. We aimed to highlight the possibility of establishing individualized treatment protocols based on these parameters, which are easy to determine from samples of the patient’s peripheral blood by using cheap commercial kits.

We have established that mortality rates at three months following the surgical intervention were 85.7% in patients presenting with KB values between 0.2 and 0.5 mM, and 100% in patients with KB values above 0.5 mM (Table 3). Additionally, we want to highlight that KB values over 0.2 mM were recorded only in patients with GBM, suggesting the fact that this aggressive, heterogeneous tumor may benefit from an extremely complex adaptive metabolic mechanism, and dietary changes or medication administration for reaching the “metabolic management area” could be ineffective in this category of patients [6,33,36,54]. In contrast, KB values of less than 0.2 mM were recorded in patients with ASTRO G4, which were similar values to healthy subjects.

Most studies that present the use of ketogenic diets as adjuvant therapies in glioma patients do not take into consideration the histopathological classification, therefore, the results are often contradictory [5,30,62]. 

Sargaço et al. tried to establish the effects of ketogenic diets in patients with gliomas in a systematic review. They found nine relevant studies showing an overall survival increase (in half of the analyzed studies), as well as the quality of life (in 25% of cases), in patients who were administered ketogenic diets, and only in one quarter of the cases the quality of life decreased [63]. Unfortunately, the analyzed studies established the histopathological diagnosis of GBM or astrocytoma grade 2, 3, or 4 without mentioning the molecular subtype or other histopathological details of the included cases. Based on our results, we want to highlight the need of viewing GBM and ASTRO G4 as two distinct pathologies characterized by their tumoral heterogeneity.

Sperry et al. demonstrated that U87 glioma cell lines as well as cell culture lines derived from GBM patients, including those with mutations in the mTOR/AKT/PI3K/IDH1 signaling pathways, can use KBs for tumor growth under standard and physiological culture conditions. Furthermore, it has been shown that the administration of ketogenic diets to tumor-bearing animals does not decrease the rate of tumor growth or improve the survival of these animals, proving the metabolic plasticity of GBM [29]. The drawback of this study is the lack of correlation of the obtained results with the blood values of KBs.

Steroids, most commonly dexamethasone, are a standard treatment for GBM and ASTRO G4 and are administered both before and after the surgical intervention and during chemotherapy/radiotherapy. The goal is to reduce the perilesional vasogenic edema as well as to prevent and even treat increased intracranial pressure. However, steroid administration is associated with a multitude of side effects, such as abnormalities in glucose metabolism, gastrointestinal complications, myopathies, insomnia, and anxiety. Although most complications are reversible after treatment discontinuation, 50% of patients have persistent disturbances in glucose metabolism after discontinuing the treatment [23,24,29,64,65,66].

The mean blood glucose levels were higher in the tumor groups compared to the control group, secondary to dexamethasone administration (treatment was administered for 1 to 3 days prior to glucose determination), with a statistically significant difference between the two groups (*p*-value = 0.0003, Table 2). These increased values show changes in the glucose metabolism, which in turn lead to an increase in the fuel needed for tumor development. On the other hand, the mean blood glucose level in ASTRO G4 patients was 8.9 mg/dL higher than in GBM patients. This slightly higher value can be explained by the administration of higher doses of glucocorticoids in patients with ASTRO G4 because the imaging aspects of astrocytoma grade 3/4 are characterized by peripheral low T1 signal and high T2 signal areas, sometimes without contrast enhancement on the T1C+ sequences, which can erroneously be interpreted as larger perilesional edematous areas (Figure 2d–f, red circle indicates tumor boundary). Zhou et al. analyzed 10 articles in a systematic review that included a total of 2230 patients diagnosed with GBM or ASTRO G4 and concluded that dexamethasone administration significantly decreases patient prognosis [67]. The results presented by us also raise questions about the doses and timing of dexamethasone usage throughout the course of the disease.

Another important marker calculated was GKI, designed to prove the effectiveness of various nutritional interventions which lead to lower blood sugar levels and increased KB levels. Artificial intelligence (AI) has the potential to play a significant role in evaluating markers like GKI and assessing their effectiveness in various nutritional interventions aimed at reducing blood sugar levels and increasing KB levels. Furthermore, AI can help in optimizing personalized dietary plans for individuals based on their unique metabolic responses. It can consider factors like genetics, lifestyle, and medical history to recommend tailored nutritional interventions that are more likely to achieve the desired GKI outcomes. In summary, AI can enhance our understanding of the impact of nutritional interventions on markers like GKI by efficiently analyzing complex data and providing evidence-based insights to guide dietary recommendations for individuals seeking to manage their blood sugar and ketone levels [68].

Due to the changes in blood glucose values secondary to steroid administration, GKI values were altered in the tumor groups, with a statistically significant difference between the three groups of patients (*p*-value = 0.008, CI 95%, Table 1). Although we do not expect significant changes between patients with GBM and those with ASTRO G4, the GKI value was 34.4 mm higher in the group of patients with ASTRO G4. These values emphasize the need to manage the two pathologies individually, highlighting the GBM heterogeneity. 

While our study provides valuable information, there are several limitations to consider like the relatively small number of patients and the fact that the two tumoral groups of patients had to receive dexamethasone to decrease the perilesional edema which has led to elevations in the levels of blood glucose and GKI.

## 5. Conclusions

Highlighting new markers that are feasible to acquire (such as KB and GKI) which could also become additional therapeutic targets represent important steps toward treatment individualization and survival rate prolongation in patients with GBM and ASTRO G4. 

Although this study was performed on a small group of patients, we have demonstrated statistically significant differences in the peripheral blood values of KBs and GKI between these two pathologies (GBM and ASTRO G4) and compared them to a control group; therefore, these two pathologies need to be viewed and managed as two distinct pathologies. We can also emphasize that KB values over 0.5 mM represent a negative prognostic factor in patients with GBM.

Establishing individualized adjuvant therapies based on reducing blood glucose levels and increasing KB levels in patients with ASTRO G4 could lead to survival rate improvements in this category of patients, considering that the KB values in these patients are like those of healthy subjects (below 0.2 mM). In contrast, nutritional changes may be ineffective in patients with GBM due to the heterogeneity and adaptive mechanisms of this pathology. 

Our study also raises the need for larger clinical trials which are aimed to demonstrate the benefits of dexamethasone administration in patients with GBM and ASTRO G4. 

## Figures and Tables

**Figure 1 brainsci-13-01307-f001:**
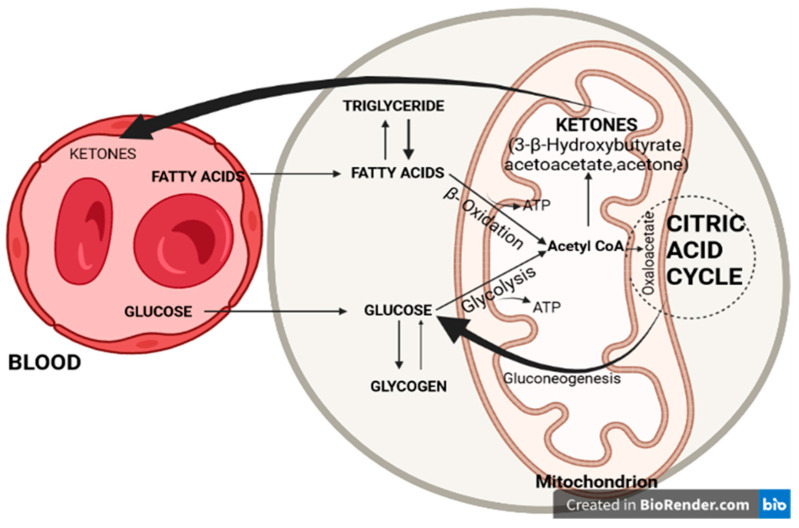
Glucose and fatty acid metabolism in hepatocyte mitochondria.

**Figure 2 brainsci-13-01307-f002:**
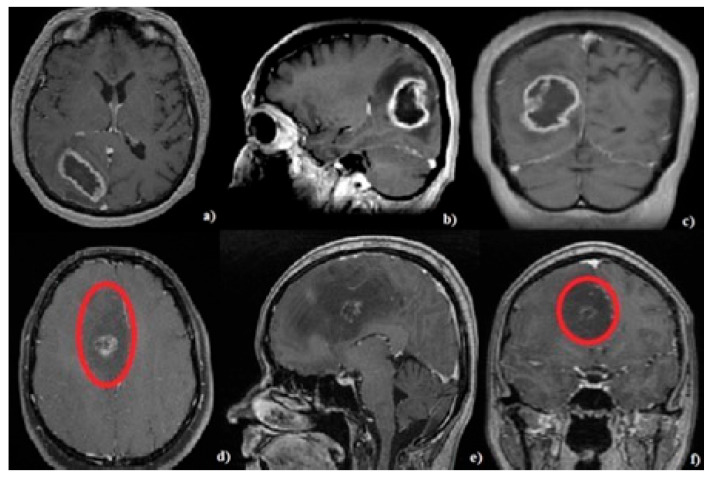
Cerebral MRI, T1C+ sequence: (**a**–**c**) showing a glioblastoma patient. (**d**–**f**) showing a grade 4 astrocytoma patient. Red circle (**d**,**f**) shows tumor boundaries.

**Figure 3 brainsci-13-01307-f003:**
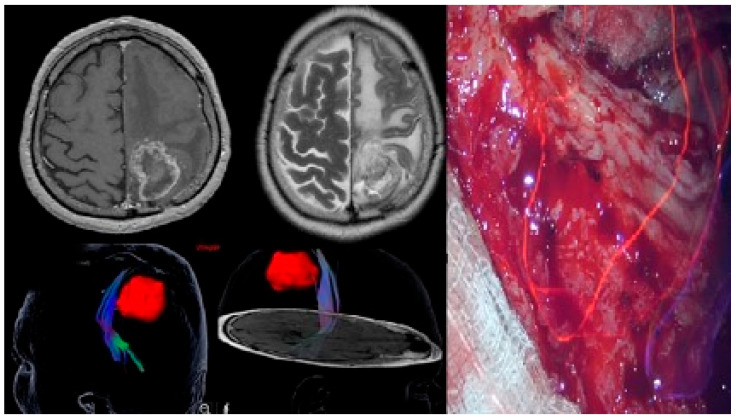
Postcentral GBM (cerebral MRI, T1C+, and T2 sequences). Tractography showing the pyramidal tracts and image injection into the surgical microscope (Captiview, Leica Microsystems GmbH 35578 Wetzlar Germany).

**Figure 4 brainsci-13-01307-f004:**
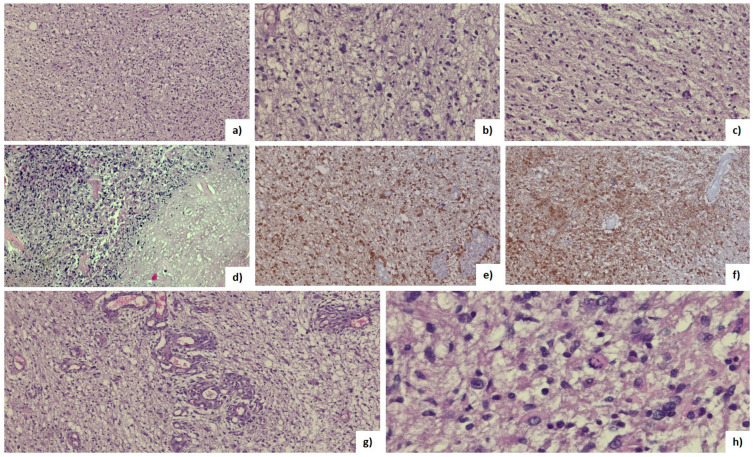
(**a**) Hematoxylin-Eosin (HE) GBM (10×); (**b**,**c**) nuclear pleomorphism (HE, 20×); (**d**) tumor and necrosis (HE, 20×); (**e**,**f**) positive IDH 1-R132H mutation on immunohistochemistry (HE, 20×); (**g**) microvascular proliferation with hyperplastic endothelia (HE, 20×); (**h**) atypical mitosis (HE, 40×).

**Table 1 brainsci-13-01307-t001:** General and specific characteristics.

Tumor Group				Standard Deviation		Control Group	*p*-Value
**General Characteristics**							
		ASTRO G4	GBM	Tumor group	Control group		
Gender % (N)	Male	66.7 (4)	56.3 (9)			59.1 (13)	
	Female	33.3 (2)	43.7 (7)			40.9 (9)	
Years	Average age	48.1	55.6			53.9	0.512
	Age range	34–66	44–78	56.09 ± 12.33	53.45 ± 14.10	36–78	
Specific Characteristics (average)							
Wight (kg)		80	89.6	85.00 ± 12.72	82.23 ± 16.70	82.2	0.538
Height (cm)		169.8	175	173.6 ± 7.681	171.2 ± 10.13	171.2	0.379
BMI		27.8	28.3	28.15 ± 3.867	28.02 ± 4.136	28.0	0.914
Ketone Bodies (mM)		0.13	0.26	0.227 ± 0.2004	0.0773 ± 0.0812	0.08	0.0061
Glycemia (mg/dL)		138.5	129.6	132.0 ± 39.43	96.73 ± 11.78	96.7	0.0003
GKI (mM)		63.7	29.3	38.68 ± 29.80	16.85 ± 21.37	18.0	0.0080
Clinical debut (weeks)		12.7	2.9	85.00 ± 12.72	82.23 ± 16.70	82.2	0.538
%	Headache	83.3	93.7				
	Motor deficit	33.3	56.3				
	Confusion	16.7	50				
	Seizures	16.7	25				
	Aphasia	16.7	12.5				

**Table 2 brainsci-13-01307-t002:** Imaging and histopathological characteristics.

		ASTRO G4 (%)	GBM (%)
Tumor location	Frontal	50	50
	Temporal	33.3	31.2
	Parietal	16.7	37.5
	Occipital	16.7	31.2
	Insular	0.0	12.5
Cerebral hemisphere	Left	33.3	37.5
	Right	66.7	62.5
Tumor size (mm)		40.1/30.2	48.4/39.6
Size of the perilesional edema (mm)		23.6/3.5	23/2.4
Ki-67 index	≤15	50	18.7
	15–30	16.7	37.5
	≥30	33.3	43.8
P53 +		83.3	50

**Table 3 brainsci-13-01307-t003:** Ketone body values and mortality at 3 months after the surgical intervention.

	Ketone Bodies	Tumor Group		Control Group (%)
		ASTRO G4 (%)	GBM (%)	
	≤0.2 mM	100	56.25	100
	>0.2 mM	0.0	43.75	0.0
MORTALITY AT 3 MONTHS	≤0.2 mM	0.0	0.0	
	0.2–0.5 mM	0.0	85.7	
	≥0.5 mM	0.0	100	

## Data Availability

MDPI Research Data Policies” at https://www.mdpi.com/ethics.

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
