# Peer review of "The Role of Ketone Bodies in Treatment Individualization of Glioblastoma Patients"

_brainsci, 2023, doi:10.3390/brainsci13091307_

Round 1

Reviewer 1 Report

The manuscript(brainsci-2598228) entitled “The role of ketone bodies in treatment individualization of glioblastoma patients” is overall interesting, logical, and presented in a easily understood manner. A few comments below:

1. One of tumoral groups is Astro G4. It would be interesting to see the ratio of IDH mutation status in this group. Did the authors investigate the correlation between IDH mutation and the levels of KB?

2. What does IGK mean in table 1? In addition, improving the formatting of table 1 and 2 is suggested.

3. In the glucose-ketone index (GKI) formula (page 4, line 135), please check if the GKI needs the unit (mM). The unit is incorrect (page 4, line 133: Mm -> mM)

Minor editing of English language required.

Author Response

  1. All Astro G4 patients were IDH mutant. We have mentioned this fact in the "General clinical features" section - line 205.
  2. IGK was a writing mistake. We have made the correction from IGK into GKI and, we have improved the formatting of table 1 and 2.
  3. We have made the correction from Mn into mM.

Thank you for your comments and observations.

Reviewer 2 Report

Interesting paper about a feasible and cheap mean that in a wider sample might be promising.

I only would suggest to the author to:

- reduce the lenght of the introduction and focus on the topic of the paper 

- specify in the method section if the sampling are performed in all admitted patients in the same conditions (i.e. fasting? in the early morning of the 1 and 3 day?)

Put more short sentences replacing longer ones.

Author Response

  1. We have reduced the length of the introduction by moving some of the paragraphs into the discussion section of the article.
  2. We have mentioned the sampling method in lines 111 and 112.

Thank you for your comments and observations.

Reviewer 3 Report

As the most common and aggressive primary brain tumor in the adult population, glioblastoma is a fast-growing tumor and generally considered incurable. In this study, the authors conducted a prospective observational study to identify various markers, including the glucose levels, ketone bodies, and the glucose-ketone index in three groups of subjects, which were two tumoral groups of patients with histopathological confirmation of the tumor and a control group consisting of subjects with no known pathological history. The obtained results revealed statistically significant differences between the mean values of glycemia,

ketones, and glucose-ketone index between the groups of patients. This is an interesting study that can add to the body of our knowledge on a crucial topic, with potential translational benefit to improve diagnostics and prognostics. However, there are some comments to help the authors improve the quality of their manuscript before it is considered further for publication. Below, I provide detailed comments:

 1.     Abstract: Some more information on basic patient characteristics should be provided, including the number of patients in each group, mean age, sex, etc.

2.     Abstract: Line 23: To improve the readability, please change “We have conducted a prospective observational study in which we have identified…” to “We have conducted a prospective observational study to identify…”

3.     Abstract: Line 27: “There were statistically significant differences…” Please provide some actual results.

4.     Abstract: 0%: Please provide the number to more decimal places.

5.     Abstract: Change “cheap and easy” to “feasible” to improve the readability.

6.     Abstract: Change “treatment individualization” to “personalized medicine”

7.     Figure 1 resolution needs some improvements.

8.     Introduction: The first paragraph is well-written and to the point.

9.     Introduction: Line 46: Change “old” to “previous”

10.  Introduction: Line 47: Change “uses” to “used”

11.  Introduction: Line 66: To improve the readability, change “Many recent studies have questioned about the fact that in addition to glucose, GBM may also use other fuels such as ketone bodies (KBs) to generate energy…” to “Many recent studies have questioned the possibility of using other energy sources, such as ketone bodies (KBs) by glioblastoma to generate energy."

12.  Introduction: Line 67: The authors have abbreviated glioblastoma before to GBM, so to make the text consistent, make sure GBM is used throughout. Please check the entire text.

13.  Introduction: references are clustered and used at the end of each paragraph. Please make sure to distribute the references to sentences inside the paragraph.

14.  Introduction: “However, the average survival duration is still less than 15 months, and the 5-year survival is below 10%”. This is important, and to enrich the Introduction, the authors can mention about the wider impact of glioblastoma diagnosis, for example, discussing that patients and survivors of GBM are at a higher risk of suicidal ideation and attempt. Please consider citing the following recent systematic review: Suicidal ideation and attempts in brain tumor patients and survivors: A systematic review. Neurooncol Adv. 2023 May 12;5(1):vdad058. doi: 10.1093/noajnl/vdad058. PMID: 37313501

15.  Materials and methods: Line 103: Change “commercial kits” to “commercially available kits”

16.  Materials and methods: were patients blinded to the study, so they were not aware of the study aims?

17.  Materials and methods: This should be removed: “and to simultaneously highlight the need to administer adjuvant therapies such as ketogenic diets according to these parameters, therefore individualizing treatment.” This is not necessarily the aim of the paper, as it has wider implications. It is understandable that such findings can be used for such purposes, but it was not the aim of the study.

18.  Materials and methods: Line 124: “known personal pathological history” Can you please provide some examples?

19.  Line 128: “commercial kits” Please provide details, such as company name, product number, etc, so other colleagues around the world can replicate and expand the findings.

20.  Line 141: “Karnofski Performance Score” should be abbreviated to KGS when it first times appears in Line 123.

21.  Line 146: “Follow-up MRIs were performed every 3 months.” For how long? Please specify.

22.  Discussion: Line 265: Change “seems to” to “can”

23.  Line 280: Change “we want to emphasize the hypothesis according to which” to “we explored the possibility of ”

24.  Line 282: “We aimed to highlight the possibility of establishing individualized treatment protocols based on these parameters”. This is an interesting point to discuss. To enrich the manuscript and open the discussion, the authors can add a few sentences highlighting that in future large-scale studies, artificial intelligence can be used to build diagnostic and prognostic tools to predict GBM patient outcomes. Please consider citing the following paper on the usage of AI in neurosurgery.

Neurosurgery and artificial intelligence. AIMS Neurosci. 2021 Aug 6;8(4):477-495. doi: 10.3934/Neuroscience.2021025. PMID: 34877400

25.  Discussion: Line 313: “Steroids, most commonly dexamethasone…” How can this affect the findings from the current study? Please discuss this and mention it as a limitation of the study.

26.  Would it be better to have another group of patients without brain tumors but another pathology, such as autoimmune disease, who were using steroids and compare the outcomes? Please discuss.

27.  A paragraph before conclusion should be added to discuss the limitations of the study (low sample size, steroid usage, etc)

28.  Conclusion: Change “cheap and easy to determine” to “feasible to acquire”

29.  Conclusion: Line 356: Change “say” to “emphasize”

I have provided detailed comments in the previous section, which include minor changes to the language to improve the readibility.

Author Response

Thank you for your observations and comments. We modified the text according to your observations and included the two bibliographic notes into the main text.

  1. The patients knew about the aims of the study and provided informed consent which is mentioned in lines 98 and 99.
  2. We have included some ideas regarding artificial intelligence in lines 353-362.

25/27. Dexamethasone administration increases the serum glucose levels and so the GKI is influenced. We have written a new paragraph in which we have discussed the limitations of the study in lines 369-372.